# Clinical Leadership and Management Perceptions of Inpatients with Obesity: An Interpretative Phenomenological Analysis

**DOI:** 10.3390/ijerph17218123

**Published:** 2020-11-03

**Authors:** Danielle Hitch, Fiona Pazsa, Alison Qvist

**Affiliations:** 1Allied Health, Western Health, 3021 St. Albans, Australia; fiona.pazsa@wh.org.au; 2Occupational Therapy, Deakin University, 3217 Geelong, Australia

**Keywords:** attitudes, clinicians, hospital, in-patient, obesity, perceptions

## Abstract

While obesity is recognized as a key global public health issue, there has been no research to date on the perceptions of care for people with this condition held by individuals in positions of organizational power. The aim of this study was therefore to describe the perceptions and experiences of clinical leaders and managers of providing care to inpatients with obesity at a metropolitan public health service. This study applied an interpretative phenomenological analysis (IPA) approach to qualitative research, conducting interviews with 17 participants. Their perceptions of care for inpatients with obesity encompassed both their personal understanding as an individual, and their observations about the organizational, patient and carer perspectives. Three overall themes were identified: (1) the problem of inpatients with obesity, (2) inpatients with obesity as sources of risk and (3) personal and professional perceptions of inpatients with obesity. While clinical leaders and managers were aware of the potential impact of stigma and weight bias on care given to this cohort, elements of implicit bias, stereotyping, “othering” and ambivalence were frequently present in the data. Ongoing efforts to improve care for patients with obesity must therefore include efforts to address perceptions and attitudes at all organisational levels of the workforce.

## 1. Introduction

Obesity is a growing public health issue around the world, with rates of this condition in the population steadily increasing over the past decade [1]. While the consistent identification of patients with obesity can be challenging in hospital based health settings [2,3], this cohort is known to incur medical costs approximately 30% higher than patients of normal weight [4]. The efficient and effective treatment of inpatients with obesity is therefore a priority for clinical leaders and managers, as part of their overall stewardship of hospital services.

While national policies are now prioritizing the prevention of obesity across the population [5], people with obesity are likely to comprise a significant proportion of the inpatient population for some time to come. Recent data indicates that 63% of the Australian adult population meets the criteria for obesity or overweight, which is the fifth highest prevalence amongst Organisation for Economic Cooperation and Development (OECD) member countries [6]. International research has confirmed that inpatients with obesity are more likely to experience lower quality care, increased length of stay and adverse events such as the development of pneumonia, infection, falls, and pressure injuries [7]. The perceptions of those in clinical leadership and management within hospital services are therefore important to understand, in regards to both current practices and future developments.

### 1.1. Weight Bias in Hospital Services

Weight bias (also known as “weight stigma”) is a broad concept which comprises three conceptually independent constructs: prejudice, stereotyping and discrimination [8]. Prejudice arises from negative perceptions of patients based significantly on their weight, while stereotyping is based upon personal views regarding the factors that cause and maintain obesity. Discrimination occurs when people take action on the basis of their prejudice or stereotypes, and one of the challenges of this phenomenon is its basis in multiple attitudes, beliefs, implicit and explicit biases [9]. Lee et al. [8] suggest that weight bias may be one of the last remaining socially acceptable forms of discrimination.

Several studies have found weight bias to be prevalent in hospital services internationally, with prevalence of up to 66% reported in some healthcare settings [10]. Prejudices and stereotypes reported to be held by healthcare workers include perceptions that people with obesity are lazy, non-compliant with treatment, ugly, unattractive, undisciplined and a frustrating group to work with [11,12,13]. Women with obesity are more likely to experience weight bias in health care [14], and people who experience this form of discrimination are more likely to avoid or disengage from the care they need [15]. Despite these negative consequences, few interventions have been developed to reduce the impact of weight bias amongst clinicians. Many of these interventions target the reduction of negative perceptions in healthcare students, however initiatives for clinicians are beginning to emerge (including innovative approaches such as simulation) [16,17].

### 1.2. Perceptions of Providing Care for Inpatients with Obesity

Despite the increasing prevalence of obesity in inpatient populations, relatively little research has been conducted into clinician perceptions of their care and management to date. Internationally, a survey of healthcare professionals across eleven countries [18] found that most agreed that obesity was a disease, however fewer than half acknowledged that genetic factors may contribute to weight gain. In regards to care, many healthcare professionals assumed that people with obesity were not interested in addressing their obesity and few felt they had the time to initiate obesity interventions with patients in their usual role.

The majority of existing studies were undertaken with nursing staff, and highlight the impact of social norms and beliefs around obesity. A study of nurses working an Australian metropolitan private hospital found some ambivalence in attitudes towards obesity (particularly between nursing and medical discourses), and the place of personal responsibility for the patient [19]. An ethnographic study of United Kingdom nursing students also described their engagement in deliberate acts of compassion to compensate from poor care from colleagues, and found a close relationship between their personal values about obesity and their provision of care [20]. The values held by German nurses around personal responsibility [21] were also found to have a relationship with reported discriminative practice towards inpatient with obesity.

There have also been two studies of nursing staff perceptions of inpatients with obesity admitted to intensive care units. A study of Canadian nursing staff in an intensive care unit found the patients with obesity were subjected to “othering”, although participants attempted to distance themselves from instances of stigma and discrimination from their colleagues [22]. Norwegian nurses also reported a sense of simultaneously experiencing negative beliefs and attitudes, while striving to provide good and equitable care to all patients regardless of their weight [23].

### 1.3. Rationale for the Study

The evidence base around clinician perceptions of providing care for inpatients with obesity is currently very limited, and largely confined to a single discipline. Patients with obesity receive care from a wide range of healthcare professionals during hospital admissions, and therefore a multidisciplinary approach to this topic is required. Previous studies have also focused on the perceptions of clinicians who directly provide care to patients; however, the review of clinical leaders and managers remains unknown. Due to their organizational positions, these people have the potential to make a significant impact on the care received by inpatients with obesity across multiple service areas. Their perceptions of both current practices and future development are therefore important to understand, as their influence is likely to have a far broader reach.

The aim of this study was therefore to describe the perceptions and experiences of clinical leaders and managers of providing care to inpatients with obesity at a metropolitan public health service. The objective of the study was to drive service improvement, and ensure best care for people with obesity receiving inpatient care at the health service.

## 2. Materials and Methods

### 2.1. Terminology and Language

People with obesity attend hospital services for a range of reasons, many of which are not directly related to their weight or size. Bariatrics is the medical field that manages obesity, however relatively few Australian public health services provide specialist bariatric assessment or intervention [24]. Despite this, participants in this study often used the term ‘bariatric’ when discussing care for inpatients with obesity. This term has been retained in direct quotes from participants for credibility, however this study utilises person first language wherever possible to promote stigma reduction and patient dignity [25].

### 2.2. Study Context

This study was conducted in a public health service, located in metropolitan Australia. The local community is located in one of the fastest growing regions in Australia, and has a higher than average prevalence of mental health issues, diabetes and obesity [26]. The local community it services has a high degree of cultural diversity and includes many low socioeconomic areas [27]. The health service includes three acute hospital campuses, along with outpatient and community services [28].

The health service previously identified a need to improve care for inpatients with obesity, on the basis of a service audit conducted in 2018/2019. The audit identified that only 2% of hospital admissions during this time were coded for obesity, despite height and weight data indicating that up to 40% of patients met the criteria for this condition. The average length of stay for inpatients with obesity was 3.43 days in acute areas (compared to 1.52 for other patients), and 24.56 days in sub-acute areas (compared to 19.57 for other patients). Audit data also showed that inpatients with obesity tended to have admissions to multiple medical specialties and venues across the health service (i.e., inpatient, outpatient, and community). This study was conducted alongside other research at the service investigating the patient experience of their inpatient care and the co-production of a new care model, whose findings are reported elsewhere.

### 2.3. Methodological Approach

The study applied an interpretative phenomenological analysis (IPA) approach to qualitative research, to describe how the participants made sense of their experiences of providing care to inpatients with obesity [29]. IPA has been used extensively within healthcare research over the past two decades, and draws upon the theoretical concepts of phenomenology, hermeneutics and idiography [30]. This methodology has been used in several previous studies of the experience of people with obesity, but this is a novel use of the method with healthcare providers for this topic.

### 2.4. Researcher Stance

A methodological review by Brown et al. [31] of studies with and about people with obesity highlighted that researcher reflexivity (including personal beliefs about obesity) was often underreported in the evidence base. All researchers completed the Attitudes Towards Obese Persons and Beliefs About Obese Persons Scales [32] as a stimulus for iterative reflection on their personal stance in relation to this study.

The first author (D.H.) is an occupational therapist and academic, with many years of experience in multiple research methods. She was the allied health research and translation lead at the health service, and had a particular interest in the impact of clinician perceptions on the quality of care provided. D.H. held predominantly positive attitudes towards and beliefs about people with obesity, and is a person with multiple chronic health conditions, including obesity.

The second author (A.Q.) is a Dietitian and Clinical Informatics Analyst. A.Q. lead the previously referenced audit within the organization and at the time of this study was undertaking a project role focused on improving processes within the organization to deliver best care for patients with obesity. A.Q. has a strong belief in the environmental impacts on obesity, which includes access to equitable, appropriate and dignified healthcare.

The third author (F.P.) is a Physiotherapist with an interest in working with patients with obesity, and is also completing projects in this field that will contribute to best care in this patient cohort. F.P. lives a healthy lifestyle and encourages others to do the same, however understands the challenges faced by people that lead to obesity.

### 2.5. Information Power

The Model of Information Power [33] provided a framework for both sample size scoping and subsequent data analysis. The model indicates the more information held within the sample that is specifically relevant to the study aims, the smaller the sample size needed to gain a rigorous overall understanding while accommodating diversity. While the study aim is relatively broad, the explicit focus on a single health service bounds its scope more tightly. The model also states that samples must include people with specific and relevant experiences, knowledge or properties, and Malterud et al. [33] notes that samples from populations not previously studied hold greater information power. All potential participants with relevant experience of the specific phenomenon within this specific service setting were invited to participate.

The use of semi-structured interviews to collect data addresses another aspect of the model, by promoting the use of quality interview dialogue. The interview prompts were based upon the existing evidence base, and both interviewers were experienced clinicians with advanced skills in interviewing. Finally, the iterative nature of the IPA analysis strategy and interrogation of data against existing research clearly located the findings into their knowledge context. The research team acknowledges the study findings may not reach the traditional definition of saturation, and is limited in this pursuit by its restriction to a single health service.

### 2.6. Recruitment and Data Collection

Purposive, convenience sampling was adopted, which sought to recruit clinical leaders and managers who had worked with teams that care for patients with obesity in the past 12 months. Potential participants were identified from the membership of a Bariatric Working Party, supplemented by additional key stakeholders nominated by the researchers based on their knowledge of the service. The focus of this working party was patients with Class 3 obesity (also known as morbid obesity), whose weight exceeds (or appears to exceed) the identified safe working load/weight capacity of standard hospital equipment OR size restricts the use of standard furniture, mobility or functional level. An invitation to participate (including a plain language statement) was distributed via email by the first author who had no pre-existing relationship with study participants. Potential participants expressed their interest by contacting the first author to arrange a mutually convenient time and venue for interview. Written, specific consent was collected from each participant prior to their interview commencing. The study was approved by the Organisations Low Risk Ethics Panel (HREC/18/WH/47094: 28/12/2018).

All interviews were undertaken by D.H. and A.Q. between February and May 2019, and occurred within working hours at the health service. Each interview was guided by a bespoke set of prompts, developed from the findings of a previously undertaken literature review and the clinical experience of the researchers. Prior to use, research colleagues outside of the study team were approached to provide consultation and feedback on the interview prompts. The interviews were semi structured, which allowed for flexible administration and enabled participants to share and expand on their experiences as ‘experiential experts’. All interviews were digitally recorded for verbatim transcription, with occurred within 2–4 weeks and included de-identification of all data through the use of an alphabetical identifier (i.e., Participant A, Participant B). No field notes were taken during the interviews to allow the interviewers to be completely focused on the open-ended style of interviewing. Interviews lasted between 25 and 64 min. Transcripts were then returned to all participants for member checking, with only one transcript returned with grammatical and spelling corrections.

A total of 17 out of the 33 potential participants invited (52%) consented to participate in this study. Participants included representatives of physiotherapy, occupational therapy, nursing and occupational health and safety practitioners.

### 2.7. Data Analysis

IPA began with ideographic analysis, undertaken individually for each transcript using the Dedoose platform [34]. One researcher (D.H. or A.Q.) analysed half of the transcripts each, producing codes on a line-by-line basis to indicate their interpretation of the meaning of that text. As per the IPA method, these codes expressed the researchers attempt to capture how the participants made sense of their personal and social world, in relation to the care of inpatients with obesity [29]. A non-coding researcher then reviewed the analysis each transcript, for both coherence and consistency of code application. The few instances of disagreement identified were resolved during regular discussions amongst the research team, which also included preliminary consideration of the relevant intra-interview themes identified within the ideographic analysis.

All researchers (D.H., A.Q. and F.P.) then met to undertake the next phase of the IPA method, which is known as inductive analysis. All codes and intra-interview themes were printed onto notes, which were then physically re-arranged on a table into inter-interview or overall themes via discussion and iterative reflection. Five overarching themes were identified, one of which (perceptions of inpatients with obesity) is the focus of this paper. Findings related to the other four themes, which were all related to service and health system factors, are reported together in a separate article. Results of the inductive analysis were reported back to participants via email.

The final phase of IPA analysis is referred to as interrogative, and involves the comparison of the ideographic and inductive analyses with other research from the field [35]. Two researchers (D.H., A.Q.) completed a further literature review to discover evidence specifically relevant to the findings. This comparative analysis forms the basis of the discussion section of this article. During the consolidation of the three phases of analysis, the Consolidated Criteria for Reporting Qualitative Research (COREQ) was utilised to maximise the quality of presentation for the study [36].

### 2.8. Trustworthiness

Trustworthiness refers to the efforts made by qualitative researchers to increased rigour, by addressing dependability, credibility, transferability and confirmability [37]. In this study, multiple measures were taken to ensure trustworthiness. Member checking was undertaken with all participants, and a plan language summary of the overall findings was also provided at its conclusion. One researcher coded all transcripts, before they were independently reviewed by a second researcher and discussed within the team to reach a shared interpretation of the overall themes. The reflective task completed by researchers around their personal stance on obesity began a process of iterative reflection and mutual peer review that continued throughout the study. The first author (D.H.) is an experienced researcher, and she provided supervision and development to the other researchers around the use of IPA and general approaches to qualitative research. All study documentation was stored securely at the health service, and is available for audit on request.

## 3. Results

The perceptions of clinical leaders and managers of care for inpatients with obesity encompassed both their personal understandings as an individual, and their observations about the organisational, patient and carer perspectives. Three interdependent overall themes were identified in the data: (1) the problem of inpatients with obesity, (2) inpatients with obesity as sources of risk and (3) personal and professional perceptions of inpatients with obesity.

### 3.1. The Problem of Inpatients with Obesity

Participants most frequently discussed the challenges and problems associated with caring for inpatients with obesity, often conveying a sense of irritation and negativity; “We’ve joked about we just need a bariatric ward but who would want to work on a bariatric ward?” (G). While some participants expressed pride in the standard of care provided for inpatients with obesity by their service, perceptions of inpatients with obesity as “tricky” were prevalent throughout all stages of their admissions, and began influencing care even prior to admission; “Their weight is often the first thing that is said or thrown out to make sure that you have appropriate equipment, so they come already with this persona that they are hard work” (A).

Many examples of staff disengagement or distancing from inpatients with obesity were recounted. Sometimes, these behaviors were attributed to a belief that treating these patients was beyond their current clinical knowledge or skills; “Everybody knows what to do with a pouch patient whereas a bariatric patient comes in and everyone just goes, Oh” (F). However, a reluctance to provide care for inpatients with obesity was also thought to be influenced by the pressures of competing demands; “No it’s too much work, I don’t have time or I’m not bothering. Somebody else can shower or somebody else on the next shift can do that” (A). These attitudes were also reported to existing in higher organisational levels, although they were not thought to be prevalent throughout the health service; “I have spoken to senior people who have said to me we just don’t think we should have to look after these people they should have to go to another hospital or another facility” (F). A sense of inpatients with obesity as a separate group or ‘others’ was also evident in the language observed by some participants, which focused on size as a defining feature of identity; “I don’t want the staff’s faces to fall when they hear they’re getting a bariatric” (J).

However, some participants attempted to reframe their work with inpatients with obesity more positively; “We shouldn’t look at it as a burden, we should look at it as a new challenge” (N). The locus of responsibility for challenges was also questioned by some, who recognized a tendency to attribute the perceived problems to the patients themselves; “They’re often lying on the bed and being spoken about like they’re a problem, even though it’s the bed that’s the problem” (J). Despite the prevailing perception of inpatients with obesity as problematic, some participants also reflected the diagnosis was not an automatic indicator of their individual circumstances, capabilities and capacities; “they’re literally walking past you after having a shower and you say to the nurse ‘why did you want us to come help, is there an issue?’ ‘Oh no, we just looked at the weight’” (D).

### 3.2. Inpatients with Obesity as Sources of Risk

Risk was a major recurring theme, and exerted a significant influence on participant perceptions of inpatients with obesity. This group of patients were perceived to pose a much higher risk of both critical incidents and complications; “The risk is greater because they are unpredictable… you probably have to be more conservative” (E). Common areas of increased risk identified when caring for inpatients with obesity included manual handling, wound care, access issues, miscommunication, podiatric issues, compromised cardiovascular function, insufficient staffing, inappropriate positioning, the ageing workforce and physical burnout from the physical care effort required. These risks were identified for both clinicians (particularly in regard to increase physical exertion), and also patients whose recovery and participation in care may be compromised by limitations related to their physical and medical complexity. The risks identified for clinicians were not solely physical, as the perceived additional demands of caring for these patients also believed to do psychological harm; “I had to support the nurses actually psychologically, because their greatest fear was if they got injured with manually handling these patients” (N). However, varying opinions between clinicians, patients and carers around what constituted acceptable risk were also observed, which needed to be negotiated as part of care; “I think some families are willing to do more than we can say is safe… it’s about defining risk” (E).

Unmanaged risks were seen as an important threat to providing quality care, given that clinicians need to remain safe and functional for sustainable work performance. Some participants believed that advocating for clinician safety also supported the needs of inpatients with obesity, given the existence of shared goals; “We have to be ok with saying I’m not prepared to do x unless you get me the right piece of equipment as well which is for the patients sake as much as for the clinicians” (C). The impact of potential risk was also perceived as reaching beyond the inpatient with obesity and their treating clinicians, given the potential impact on other patients in the ward; “It places at risk everybody else that’s not got somebody assigned to them, while you were all in there caring for the one person” (J). An example of this broader impact was provided in discussions about emerging complacency for routine care, stemming from hypervigilance around the need to manage the risks presented by inpatients with obesity; “We’re actually starting to see a bit of a trend now when we’re getting injuries now between the 60–80 kg ranges, people are a little bit complacent. Whereas now when people see a bariatric patient they do take on more controls to control the risk …” (K).

However, a key tension was identified between the need to be protective and the imperative to provide personalized care. Some participants described taking uncomfortable risks with inpatients with obesity, due to service pressures or a desire to be client centred; “There was lots of manual handling risk that yes I put myself in, but I was trying my best to support the patient and the family” (A). In contrast, other participants observed practices where the blanket application of clinical policies led to a poor response to individual needs. In the following example, an inpatient with obesity was mobilised for a scan to gain an accurate weight measurement.

“I said do not get this man out of bed and you know I couldn’t have been more firm, I just thought this is just so dangerous. Once he gets out he’s not going to be able to get back in… and it’s going to take 8–10 people to get him off the floor and it’s going to be humiliating for him… we not looking at that whole picture, we’re just looking at that tiny little (issue)” (F).

The complex needs of inpatients with obesity was also perceived to be a poor fit with the specialized and fragmented structure of many health systems, introducing risk when a specialist (rather than holistic) approach was adopted; “I think everything was overlooked in regards to his cardiac history, his respiratory history, and they just focused on the cellulitis” (H).

Despite the perceived pervasiveness of risk, the clinical leaders and managers generally felt confident about supporting themselves and their peers to prioritise personal health and safety when required. Several organisational supports (including a no lift policy) were highlighted as enablers for protecting the health and wellbeing of clinicians; “I think we’re lucky here… we’ve got excellent Occ Health and Safety leads within the hospital who are very focused on it” (P). While the availability of suitable equipment was also identified as a key risk management strategy, it wasn’t always possible to utilize this due to a lack of availability; “They haven’t even got the patient a chair, they just sit them on a foot stool and the patient on that day was 285 kg and the foot stool was bowing, just actually bending and could have snapped” (K).

### 3.3. Personal and Professional Perceptions of Inpatients with Obesity

Underlying the participants general perceptions of these patients as risky and problematic were their personal and professional values and attitudes towards obesity. All participants identified a steady increase in inpatients with obesity being seen by the service, as obesity becomes more prevalent in Australia; “This is kind of going to be the future as well, it’s only going to increase” (A). Despite its common usage in clinical discussions, many participants expressed ambivalence about the term ‘bariatric’. While it could be supportive of identifying the specific needs of these patients, it was also believed to potentially contribute to stigma; “As soon as they are above a certain weight, they are “bariatric”… we don’t say label someone that is below that and they get followed from ED to whichever ward they end up on as ‘skinny’” (A).

Participant’s general perceived obesity to be a multifaceted and complex issue, and acknowledged that the health systems itself may compound the challenges faced by these patients; “There’s all the psychosocial challenges, let’s be frank, there’s the challenges of the health system and there’s the challenges for the patient and there’s the ramifications for their families and their careers” (E). Obesity was also perceived to be a long term, slowly developing problem, and therefore multiple determinants and influences could have had an impact on the patient’s current presentation; “it’s not like they have put on 80 kg overnight… There needs to be some onus on self and what people have done to try to address the issue, but to think that it’s as simple as the food you put in your mouth is quite naïve” (C). However, some instances of more simplistic perceptions were reported from other clinicians at the service; “You do hear people saying, “Oh, that big patient” or “why don’t they just do something about it?” (O). There were also a small number of instances where a laissez faire attitude towards inpatients with obesity was described; “they can choose to live the lifestyle they live… more power to them” (J). While still present to some degree, the participants generally described simplistic or disengaged attitudes toward providing inpatient care for patients with obesity as on the wane, which they believed was due to an increased organizational focus on the issue over the preceding year; “Previously we probably just thought, ‘Oh, they’ll be all right. They’re only gonna be on the shower chair for 10 min,’ and wouldn’t have looked at a bigger one. They’ve got bits sticking out of the side and the back” (J).

However, participants also consistently described inpatients with obesity as passive participants in their care. Inpatients with obesity were usually described using negative terms, such as ‘demoralised’, ‘embarrassed’, ‘restricted’, ‘unstable (medically)’, ‘vulnerable’, ‘challenging’, ‘risky’, ‘shame’, ‘indignity’ and ‘isolated’. Other commonly reported responses to inpatients with obesity included fear, anxiety and sadness; “Every day I see sad stories, I just think it’s not fair, they’ve got enough challenges without you know the physical, psychological, things as well” (F). The following example illustrates the most commonly described perception of inpatients with obesity, which included elements of empathy, compassion and pity; “I think if I were in that situation I probably wouldn’t advocate to get the stuff I need because I would be embarrassed, so I would just stay on the bed that was too narrow for me, not sleeping properly because I was scared I was going to fall off the side of it” (C).

Participants also appeared to assume that inpatients with obesity were fully aware of the challenges they present to health services; “I think they know they are more difficult to handle, that there are more people required to help them, and that there is an increased risk when other people are helping them” (G). These patients were observed as experiencing embarrassment and distress at times during their admission, which participants believed resulted from their own self-concept of their presentation; “When we don’t have equipment for them it’s quite embarrassing for them, when they are being squeezed in to beds or squeezed in to chairs” (N). Following on from this, inpatients with obesity were also often assumed to accept personal responsibility for any negative care outcomes that resulted from their size; “How would you feel if you were the bariatric person and a staff member hurt themselves looking after you? … You’re already conscious, generally, that you’re creating extra work. You would feel terribly” (J). As the ‘blame’ for poor care was believed to reside with the patient, participants also frequently indicated they believed these inpatients are accepting of less than optimal care; “Cause there just feel like they don’t have the right to ask for anything and they just sit quietly in the corner most of the time and just grin and bear it” (F).

Participants clearly acknowledged that clinician perceptions have the potential to impact upon on the quality of care provided to inpatients with obesity. Many believed these patients delay engagement with health services due to poor experiences or the fear of stigma, which contributes to more advanced disease and complex problems; “They kind of become really quite unwell then they present… they put off seeking treatment and it’s quite sad when you see their needs are in terrible condition” (N). During admissions, several examples were provided of inpatients with obesity suppressing requests for assistance so as not to be an inconvenience to the clinicians; “They will say ‘I won’t have a shower today because it will make it easier for them’ and they are compromising themselves really” (H). Delays in care were also frequently attributed to difficulties in obtaining suitable equipment, poor communication practices and inappropriate infrastructure in the hospital environment. As a result of these delays and less than optimal engagement in care, inpatients with obesity were broadly perceived to progress more slowly in their recovery than other patients; “These patients don’t make daily gains, they make weekly gains… and then obviously they’ve got the challenge of being stuck in hospital and that… lack of engagement… it’s sort of self-perpetuating of the cycle” (E).

The previously identified tension between providing safe care and being client centred was also evident in the participant’s personal and professional perceptions of inpatients with obesity. Participants also believed these patients were reticent about asserting their care needs for fear of being treated ‘differently’, which was acknowledged to reflect reality in some cases; “You have that embarrassment of having to take them to another ward and get them to stand on a big plate scale” (J). Their passivity was also thought to be influenced by their awareness of clinician frustrations about care delays and other clinical challenges; “They feel that they’re a problem, because everyone says, ‘Oh well, we’ve got to wait till the chair’s available,’ or ‘We’ve got to wait until this arrives or that’” (O). Some participants also highlighted a lack of agency for inpatients with obesity, asserting that the voices and choices of these patients are not being truly heard. A recurrent example were incidents when patients insisted on using standard equipment rather than bariatric equipment; “Sometimes, patients are not keen to have the equipment we’re providing, saying ‘No, no. I can fit in whatever’” (O). Health service policy stipulated the equipment be provided regardless to reduce risk to clinicians, meaning the patients personal care preferences were marginalized. The stigmatizing language used by some clinicians may contribute to this reluctance, despite the equipment ostensibly being better suited to their individual needs; “they feel terrible when you’ve got to say oh my God. I’ve got to get a big chair. I’ve got to get the big thing. I’ve got to get the big mover” (J).

There were very few positive perceptions of inpatients with obesity described in the data. Those which were identified were always in reference to perceptions of quality care provided by clinicians, which were not necessarily specific to their needs as a person with obesity; “She really indicated that there was some great moments in her care, people who kind of went above and beyond, one of the nurses on the ward made her a birthday cake and things that made her feel like a person rather than sort of a number in a bed” (L). Other positive perceptions regarding these patients were made in the context of their past, prior to their contact with the service; “Prior to that, he was playing golf, walking the golf course and living with his parents and he was doing ok in the community” (H).

Some participants also made recommendations for improving clinician perceptions of inpatients with obesity. The use of patient and carer stories was considered a key strategy to support greater understanding of their lived experience; “I think it’s using the patient story… I think the patients (can) identify some great in-depth ideas of what would more suit their needs” (E). More specifically, several participants advocated for consumer representatives to actively participate in professional education and training opportunities; “Having to get up and talk, saying you know this was done well for them but this wasn’t, you know you could improve on this” (P). This also extended to organisational mechanisms, such as the Bariatric Working Party, which at the time of this study did not include consumer representation; “That’s something that if the committee continues can be looked at, because I think that would be really essential in terms of getting a voice” (E).

## 4. Discussion

The findings of this study indicate the perceptions and experiences of clinical leaders and managers of providing care to inpatients with obesity at this metropolitan public health service were imbued with a sense of challenge and difficulty. These negative perceptions were pervasive throughout the inpatients admission, regardless of the area of the service they were treated by. The care of these patients was described as problematic, and framed by a focus on managing the perceived greater risks they posed. The participants personal and professional perceptions of inpatients with obesity included some acknowledge of the implicit biases held about these patients, while also including elements of stereotype and “othering”.

The clinical leaders and managers who participated in this study were aware of the potential impact of stigma and weight bias on the care given to inpatients with obesity. The prevalence of weight bias and stigma in health care is widely recognized, and has been found to have a significantly detrimental effect on quality of care regardless of the intentions of providers [38]. A significant link between weight bias and poorer mental health (particularly for patients with higher BMI scores) has also been identified, further increasing risks for poor health outcomes for these patients [39]. However, few studies to date have focused specifically on the acute or inpatient setting, or identified how this stigma may prevent or delay people from seeking healthcare in the first place. Previously Australian research [40] found that people with obesity demonstrated an increased use of medical services, but the influence of delayed or late access (as highlighted by the participants in this study) remains poorly understood.

Despite this awareness, the perceptions of participants in this study also included elements of implicit bias, stereotyping and “othering”. These automatic and unintentional responses were indicated in the judgments made about the passivity of inpatients with obesity, and assumptions regarding their acceptance of poor care and personal responsibility for their presentation [41]. The related phenomenon of “othering” where people with obesity are viewed as a distinctive, different group has also been described previously within the intensive care setting [22] and the broader discourse around obesity and health [42]. The prevailing practice of ‘distancing’ was also indicated by the disengaged care practices described by some participants, and their overall identification of poor care with the practices of other clinicians. Despite assertions about lingering social acceptance of weight bias [8], participants in this study still sought to distance themselves from overt examples of it. This may indicate that attitudes are beginning to change, but are yet to pervasively shift throughout the clinical workforce.

This ambivalence between stigma and inclusion is also echoed in the tension between providing safe and person centred care. While these inpatients are perceived to present greater risks to clinicians, evidence suggests the manual handling risks related to people with overweight and obesity are difficult to quantify and significantly influenced by the clinical environment and availability of appropriate equipment [43]. This appears to be at odds with the perceptions of risk being located with the patient themselves reported in this study, however the health and wellbeing of clinicians must be prioritized for sustainable care to be provided. As such, the needs of the health service remain the priority—and in this situation, providing truly patient centred care may not be possible.

Despite the focus of this study being inpatients with obesity, the findings also underscored how the classification of the person has having obesity becomes the central focus for all care provided during an admission. Their obesity may or may not be their primary reason for seeking care, but their other health and wellbeing needs can become overshadowed. Perceiving the patient only as an obese or bariatric patient reduces their identity to a single condition, and may increase the risk of poor outcomes if their needs as a whole (which may include multiple, complex and intersecting issues) are missed or downplayed.

Given emerging awareness of the impact of negative perceptions on the care of inpatients with obesity, what do these findings suggest as strategies for development and improvement? The participants in this study indicated that greater inclusion of people with obesity in training and service governance would help to address prevailing negative perceptions. Stigma reduction strategies have been successfully implemented in many health settings, and their effectiveness is supported by organisation wide approaches, interactive education and co-production with people with lived experience of obesity [44]. Clinical leaders and managers could provide leadership around the introduction of these strategies, but also need to participate in them on the basis of this study. Health services also need to develop greater capacity for person centred care, to enable better engagement between clinicians and patients. Resolution of the frequently reported difficulties in accessing specialized equipment and purpose-built environments [7,15,17] will be an important strategy to shift the balance while also maintaining safety.

This study has several limitations which impact upon the transferability of its results. While the health service is large and offers diverse services, the findings of this study are bounded by its limited geographical reach and national context. Not all clinical leaders and managers with experience of the care of inpatients with obesity chose to participate, and while no other major themes emerged from the data the research team cannot be certain that full data saturation was achieved. Ideally, two research members should have coded each transcript independently, however the inclusion of a code book and independent review of codes are also examples of best analytical practice. IPA entails the research team interpreting the lived experiences of participants, which introduces the risk of misrepresentation or misinterpretation. Finally, perceptions of social desirability may have been influential on the participants discussions of this topic, given they were discussing a potentially sensitive topic and their workplace practices.

## 5. Conclusions

Clinical leaders and managers strive to lead service changes that ensures best care is provided to all inpatients. Patients with obesity are particularly vulnerable to lower quality of care, directly relating to prevailing biases, stigma and stereotypes. This study utilized IPA to understand the lived experience of leaders and managers from a variety of professional groups of providing care to patients with obesity within a metropolitan inpatient health service. It demonstrates that despite widespread acknowledgement that the patient cohort is growing and a commitment by the organisation to improve care, patients with obesity are still perceived as problematic and sources of risk to themselves and treating staff. These perceptions of patients with obesity can result in “othering” and complacency around lower standards of care compared to the general hospital population. Ongoing efforts to improve care for patients with obesity must be predicated on an acknowledgement that the perceptions and attitudes of the workforce at all organisational levels towards this cohort is a fundamental influence on every aspect of service delivery.

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
