# Peer review of "Clinical Leadership and Management Perceptions of Inpatients with Obesity: An Interpretative Phenomenological Analysis"

_ijerph, 2020, doi:10.3390/ijerph17218123_

Round 1

Reviewer 1 Report

Thank you for giving me the opportunity to review the article. The topic is interesting, but methods presented in the manuscript are unclear. Therefore, several revisions are required before further consideration. I listed the comments below. 

Comments: 

Abstract: 

1. The authors should write about the details of participants (e.g. sex, age, professions). 

Materials and Methods: 

2. The authors should add the recruitment process of the participants.

3. The details of the ethical approval should be added (the name of institutional review board, approval number, and date).  

Discussion:

4. The limitations of the study are well addressed but it doesn't seem enought acording to the research. It might be necessary to propose a few more.

Author Response

Thank you for your comments. In reference to your comments:

Abstract: 

The authors should write about the details of participants (e.g. sex, age, professions). 

Age and gender were not collected for each participant, because this was deemed to be potentially identifying. Adding information about the professions to the abstract would take it over the word limit, and so we are unable to add this information here. However, the full breakdown of the professions is provided in the methodology and we have added more specific details here.

Materials and Methods: 

The authors should add the recruitment process of the participants.

The steps taken to recruit all participants is described on pg. 4, lines 171-181. Please indicate if there are further specific details you would like to see added to the manuscript.

The details of the ethical approval should be added (the name of institutional review board, approval number, and date).  

We have not used the name of the organisation, given the risk of potential identification for participants (as this was conducted at a single site). The approval number was in the manuscript on pg. 5, and we have added the date.

Discussion:

The limitations of the study are well addressed but it doesn't seem enough according to the research. It might be necessary to propose a few more.

Two further limitations have now been added to this section.

Reviewer 2 Report

Firstly, I have to apologize that  I am not an expert in the chosen research method - phenomenological analysis.   Secondly, I have some questions regarding study methods and materials:   1. The aim of this study was therefore to describe the perceptions and experience of clinical leaders and managers of providing care to inpatients with obesity at a metropolitan public health service. Please more specify the choice of respondents for your study - what selection criteria were used to select these 17 respondents. Do you only consider the position of the respondents, or were the age, gender and weight of the respondents taken into account?   2. The study lacks a clear definition of the patient's weight (degree of obesity) which responders consider as obese/overweight patient care?  Not only a degree of obesity but also patient age group could influence the risk of stigmatization among medical staff. 

Author Response

Thank you for your comments. In reference to your comments:

Please more specify the choice of respondents for your study - what selection criteria were used to select these 17 respondents. Do you only consider the position of the respondents, or were the age, gender and weight of the respondents taken into account?  

As stated on pg. 4 line 172, we sought participants who were in clinical leadership or management positions and who have worked with people with obesity in the past 12 months. The age, gender or weight of the participants was not collected – the first two variables were potentially identifying given that study was undertaken in a single organisation, and the final one was not considered relevant to the question (the obesity focus was on patients rather than the staff who were treating them).

The study lacks a clear definition of the patient's weight (degree of obesity) which responders consider as obese/overweight patient care?  Not only a degree of obesity but also patient age group could influence the risk of stigmatization among medical staff. 

We agree with your comment regarding patient age group as a potential influence on stigma. Participants were not asked to comments on their experience with specific patients with obesity, but rather their experience of this cohort overall. It would therefore have been difficult for them (and potentially inaccurate) to recall the age groups of all patients. As all were recruited from the Bariatric Working Party, they defined obesity according to the treatment criteria of the Bariatric Assessment Team. Additional information has now been added to section 2.6 to clarity this point.

Reviewer 3 Report

I was honored to review the manuscript entitled “Clinical leadership and management perceptions of inpatients with obesity: An interpretative phenomenological analysis” submitted to International Journal of Environmental Research and Public Health. The study presents high quality and deals with important clinical issue, such type of study is needed.  I have only few small remarks that authors should address properly.

There are only some points to correct:

 - please provide the list of abbreviations

 - please provide the number of ethical approval

- introduction and discussion section need improvement – please provide information on how your results will translate into clinical practice

- in the discussion section please provide study strong points  and study limitation section

- please correct typos

Author Response

Thank you for your comments. In reference to your comments:

Please provide the list of abbreviations

In the preparation of the manuscript we have followed the author guidelines. These state “Abbreviations should be defined in parentheses the first time they appear in the abstract, main text, and in figure or table captions and used consistently thereafter”. There are no directions about a list of abbreviations, and we would prefer to remain consistent with the guidelines.

Please provide the number of ethical approval

The number and date of ethical approval are provided on pg. 4 line 184.

Introduction and discussion section need improvement – please provide information on how your results will translate into clinical practice

We have discussed former findings about the impact of perceptions on clinical practice in several places in the introduction, particularly in 1.2. In these sections, we highlight that weight bias has been found to have a negative impact on the quality of care and equitable practice towards people with obesity. There are no other studies about inpatients that comment on the impact of these perceptions on practice currently available to reference in this section. The impact of perceptions on care is also addressed in several placed throughout the discussion, in reference to both the findings of this study and existing research. The paragraph beginning on pg. 10 line 487, also focuses on how the findings of this study may translate into clinical practice in some detail. Upon review we are unable to locate any other references or areas of the manuscript where further elaboration on these points could be provided without going significantly over the word limit.

In the discussion section please provide study strong points and study limitation section

The limitations of the study have been addressed in some detail in the paragraph beginning pg. 11, line 499. The methodological strengths of the study are addressed in section 2.8, and also comparatively in the discussion section. We would be happy to include further brief reference to other strengths of the study if you can provide guidance around what you believe it missing, however we are mindful of not going significantly over the word limit.

Please correct typos

The article has now been proof read and all typos corrected.